



**Modernizing the open-source community Noah-MP land surface model (version 5.0) with**
**enhanced modularity, interoperability, and applicability**
Cenlin He[1], Prasanth Valayamkunnath[1,5], Michael Barlage[2], Fei Chen[1], David Gochis[1], Ryan
Cabell[1], Tim Schneider[1], Roy Rasmussen[1], Guo-Yue Niu[3], Zong-Liang Yang[4], Dev Niyogi[4],
Michael Ek[1]
[1]National Center for Atmospheric Research (NCAR), Boulder, Colorado, USA
[2]NOAA Environmental Modeling Center (EMC), College Park, Maryland, USA
[3]University of Arizona, Tucson, Arizona, USA
[4]University of Texas Austin, Austin, Texas, USA
[5]Indian Institute of Science Education and Research Thiruvananthapuram, India
*Correspondence to*: Cenlin He (cenlinhe@ucar.edu)

**Abstract**
The widely-used open-source community Noah-MP land surface model (LSM) is designed for
applications ranging from uncoupled land-surface and ecohydrological process studies to coupled
numerical weather prediction and decadal global/regional climate simulations. It has been used in
many coupled community weather/climate/hydrology models. In this study, we
modernize/refactor the Noah-MP LSM by adopting modern Fortran code and data structures and
standards, which substantially enhances the model modularity, interoperability, and applicability.
The modernized Noah-MP is released as the version 5.0 (v5.0), which has five key features: (1)
enhanced modularization and interoperability by re-organizing model physics into individual
process-level Fortran module files, (2) enhanced data structure with new hierarchical data types
and optimized variable declaration and initialization structures, (3) enhanced code structure and
calling workflow by leveraging the new data structure and modularization, (4) enhanced
(descriptive and self-explanatory) model variable naming standard, and (5) enhanced driver and
interface structures to couple with host weather/climate/hydrology models. In addition, we create
a comprehensive technical documentation of the Noah-MP v5.0 and a set of model benchmark and
reference datasets. The Noah-MP v5.0 will be coupled to various weather/climate/hydrology
models in the future. Overall, the modernized Noah-MP will allow a more efficient and convenient
process for future model developments and applications.



**1. Introduction**

Land surface models (LSMs) are useful modeling tools to resolve terrestrial responses to and interactions with the atmosphere, ocean, glacier, and sea ice in the earth system. Traditionally, LSMs were thought to mainly provide lower boundary conditions to the coupled atmospheric models. However, modern LSMs have been increasingly employed as an indispensable component in the climate and weather systems to offer biogeophysical and biogeochemical insight for understanding and quantifying the impact and evolution of climate, weather, and the integrated earth environment (Blyth et al., 2021). LSMs have been widely applied to tackle many important societally relevant challenges, such as drought, flood, heat wave, water availability, agriculture, food security, wildfires, deforestation, and urbanization (Bonan and Doney, 2018).

Among many LSMs that have been developed in the past few decades, the open-source community Noah with Multi-parameterization Options (Noah-MP; Niu et al., 2011; Yang et al., 2011) is one of the most widely-used state-of-the-art LSMs. The article describing the Noah-MP model by Niu et al (2011) is *de facto* the most cited LSM paper in the last 10 years, highlighting its worldwide popular usage in the international science community. Compared to its predecessor, the Noah LSM (Chen et al., 1996, 1997; Chen and Dudhia, 2001; Ek et al., 2003), Noah-MP significantly improves known Noah limitations by employing enhanced treatments of vegetation canopy, snowpack, soil processes, groundwater, and their complex interactions as well as additional capabilities for critical land processes (e.g., crop, irrigation, tile drainage, groundwater, urban, carbon and nitrogen cycles). Another unique feature of Noah-MP is the inclusion of multiple physics options for different land processes, which allows the multi-physics model ensemble experiments for uncertainty assessment and testing competing hypotheses (Zhang et al., 2016; J. Li et al., 2020).

Noah-MP can be applied to various spatial scales spanning from point scale locally to ~100-km resolution globally, and temporal scales spanning from sub-daily to decadal time scales. Since its original development, Noah-MP has been used in many important applications, including numerical weather prediction (Suzuki and Zupanski, 2018; Ju et al., 2022), high-resolution climate modeling (Gao et al., 2017; Liu et al., 2017; Rasmussen et al., 2023), land data assimilation (Xu et al., 2021; Nie et al., 2022), drought (Arsenault et al., 2020; Niu et al., 2020; Wu et al., 2021; Abolafia-Rosenzweig et al., 2023a), wildfire (Kumar et al., 2021; Abolafia-Rosenzweig et al., 2022a, 2023b), snowpack evolution (Wrzesien et al., 2015; He et al., 2019; Jiang et al., 2020), hydrology and water resources (Cai et al., 2014; Liang et al., 2019; X. Zhang et al., 2022a; Hazra et al., 2023), crop and agricultural management (Liu et al., 2016; Ingwersen et al., 2018; Warrach-Sagi et al., 2022; Valayamkunnath et al., 2022; Zhang et al., 2020, 2023), urbanization and heat island (Xu et al., 2018; Salamanca et al., 2018; Patel et al., 2022), biogeochemical cycle (Cai et al., 2016; Brunsell et al., 2021), wind erosion (Jiang et al., 2021), wetland (Z. Zhang et al., 2022), groundwater (Barlage et al., 2015, 2021; Li et al., 2022), and landslide hazard (Zhuo et al., 2019).



Currently, Noah-MP has been implemented into many community research and operational
weather/climate/hydrology models, including the Weather Research and Forecasting model
(WRF), the Model for Prediction Across Scales (MPAS), the NOAA operational National Water
Model (NWM), the NOAA Unified Forecast System (UFS), the NASA Land Information System
(LIS), and the NCAR High-Resolution Land Data Assimilation System (HRLDAS).
Despite its popular usage in the international research and application communities, the Noah-MP
core code engine was designed 12 years ago and is outdated, and does not take advantage of
modern Fortran language architecture. It has a single lengthy (>12,000 lines) Fortran source file
lumping together all model physics with complex code and data structures using inconsistent
format and does not follow the modern Fortran code standard. This makes the Noah-MP model
code difficult for users and developers to read, modify, and test as well as to implement and apply
it to other community models. Furthermore, a lengthy code is error prone and challenging to debug.
These issues limit the further development and application of Noah-MP.
Therefore, this study is motivated to modernize (refactor) the entire Noah-MP model by adopting
modern Fortran code and data structures and standards, which substantially enhances the model
modularity, interoperability, and applicability. The base code used for refactoring is the Noah-MP
version 4.5 (released in December 2022; https://github.com/NCAR/noahmp/tree/release-v4.5-
WRF), and the refactoring effort does not change model physics. We release the
modernized/refactored Noah-MP as version 5.0 (v5.0; https://github.com/NCAR/noahmp), which
includes five key features: (1) enhanced modularization and interoperability by re-organizing
model physics into individual process-level Fortran module files, (2) enhanced data structure with
new hierarchical data types and optimized variable declaration and initialization structures, (3)
enhanced code structure and subroutine calling workflow by leveraging the new data structure and
modularization and refining code to be more concise, (4) enhanced (descriptive and self-
explanatory) model variable naming standard, and (5) enhanced driver and interface code
structures to couple with host weather/climate/hydrology models. In addition, we have created a
comprehensive technical documentation (He et al., 2023) to describe model physics and details of
the refactored Noah-MP and a set of model benchmark and reference datasets for future
comparison and assessment. Overall, the modernized open-source community Noah-MP model
(version 5.0) will allow a more efficient and convenient process for future model developments
and applications. The framework and practice in the course of refactoring the entire Noah-MP code
is also applicable to other LSMs and ESMs.
This paper reports the key features of the modernized Noah-MP v5.0 and is organized as follows.
Section 2 briefly summarizes the Noah-MP model physics with several updates since its original
development. Sections 3–7, respectively, introduce the key features of the modernized Noah-MP
in terms of enhanced model modularization, data type, code structure, variable naming, and





coupling structure with host models. Section 8 describes the model benchmarking and reference
datasets. Section 9 provides the release information of model code and technical documentation.
Section 10 concludes the paper with future model development plans.
**2. Noah-MP version 5.0 model physics**
**2.1 Noah-MP description**
Noah-MP (Niu et al., 2011) was originally developed based on the Noah LSM (Chen et al., 1996,
1997; Chen and Dudhia, 2001; Ek et al., 2003) to augment its modeling capabilities with enhanced
physical representations and treatments of dynamic vegetation, canopy interception and radiative
transfer processes, multi-layer snowpack physics, and soil and hydrological processes. The history
of model development and evolution has been described in the technical documentation (He et al.,
2023). Noah-MP is designed to simulate land surface and subsurface energy and water processes
in both uncoupled and coupled modes with atmospheric or hydrological models at sub-daily time
scale and high spatial resolution (even for point scale). This further allows the use of Noah-MP in
different hydrological, weather, and climate models for applications in a wide range of spatial and
temporal scales with proper integration in time and space.
Noah-MP divides its land grid into two sub-grid tiles, namely vegetated and non-vegetated grounds,
based on vegetation cover fraction. The biogeophysical and biogeochemical processes are treated
separately for the vegetated and bare grounds. Noah-MP adopts a "big-leaf" canopy treatment
characterized by canopy properties dependent on vegetation types. Noah-MP accounts for a
multiple-layer snowpack, where snow ice and liquid water content, density, depth, and temperature
are simulated dynamically. Noah-MP also includes multi-layer soil thermal and hydrological
processes with dynamically evolving soil temperature and water content. The vegetation, snow,
and soil components in Noah-MP are closely coupled and interacted with each other via complex
energy, water, and biochemical processes. Their detailed physical formulations and
parameterizations in Noah-MP v5.0 are described in the technical documentation (He et al., 2023).
Below, we briefly summarize the energy, water, and biochemical processes in Noah-MP v5.0.
**2.2 Noah-MP energy processes**
Noah-MP resolves energy budgets and processes separately for vegetated and non-vegetated
ground portions of each grid (Niu et al., 2011). The vegetation cover fraction, either from
observational inputs or model calculations based on leaf area index (LAI) inputs or predicted by
the dynamic vegetation module, is used to separate vegetated and bare grounds. The grid-mean
energy states and fluxes are calculated as an average of vegetated and bare ground values weighted
by vegetation cover fraction. For surface radiative processes driven by incoming shortwave and
longwave radiation (atmospheric forcing), Noah-MP simulates the radiative absorption and



scattering by the canopy and ground (soil/snow) as well as the longwave emissions by the canopy
and ground (soil/snow). The net absorbed total (shortwave and longwave) radiative flux is
balanced by precipitation advected heat flux, total surface sensible and latent heat fluxes, and
ground heat flux. The precipitation advected heat flux represents the heat flux advected from
precipitation (rain/snow) to canopy/ground due to the temperature difference between precipitation
(surface air) and canopy/ground. The total surface sensible heat includes the sensible heat from
canopy, snowpack, and soil surfaces. The total surface latent heat includes the latent heat from
snowpack sublimation, soil evaporation, canopy snow sublimation, canopy water evaporation, and
plant transpiration. The ground heat flux is the heat flux leaving the ground surface to drive
subsurface snow/soil phase change and/or temperature changes.

To model the aforementioned surface energy flux components, Noah-MP dynamically calculates
a number of key land surface properties, include ground snow cover fraction, surface roughness,
canopy and ground thermal properties, snow and soil albedo, surface emissivity, and canopy
radiative transfer. Many of these property and process calculations have multiple physics options
(see Sect. 2.6). Based on the canopy and ground energy balance, Noah-MP further solves the
temperature and phase change for canopy, snowpack, and soil. Figure 1 summarizes the key energy
processes and budget components as well as the energy balance equation in Noah-MP v5.0. Note
that the energy processes at glacier grids are treated similarly to those at 100% bare (non-vegetated)
ground grids except that the soil is replaced by glacier ice with ice-specific properties.

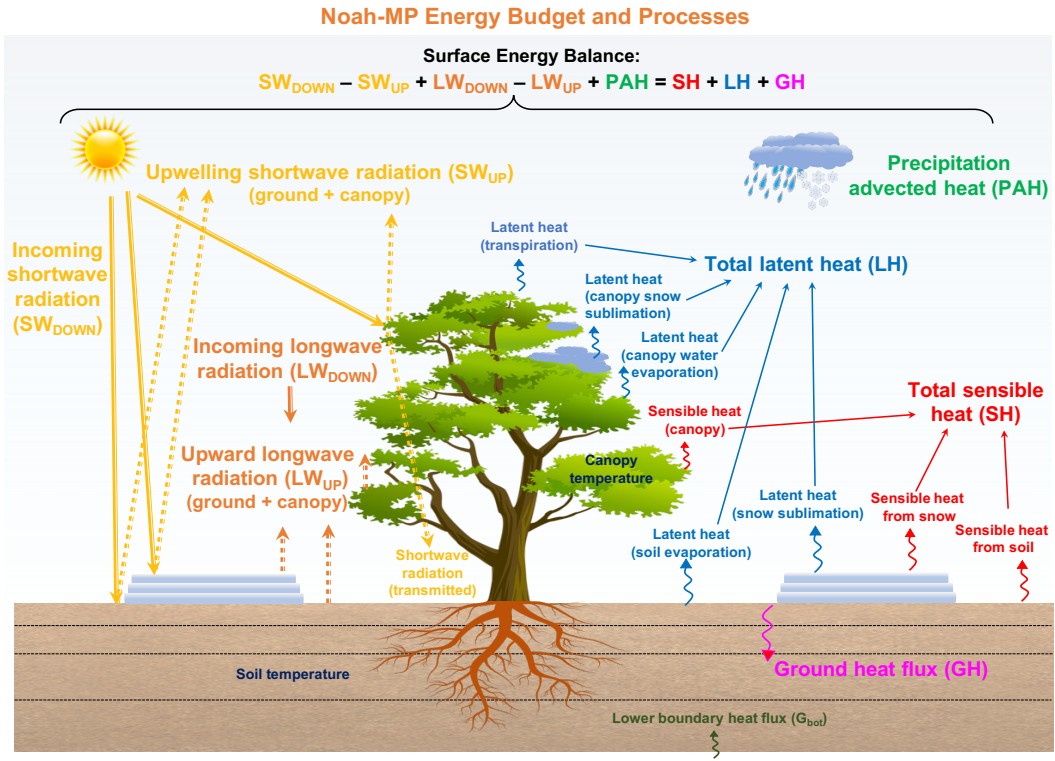

Figure 1. Schematic diagram of energy budget and processes represented in Noah-MP version 5.0.

## 2.3 Noah-MP water processes

Noah-MP accounts for five major water budget components, including precipitation, evapotranspiration (ET), total runoff, net lateral flow, and total water storage change intercepted by the canopy and in snow, soil, and aquifer. For precipitation, Noah-MP has several temperature-based rainfall-snowfall partitioning parameterizations or can use the partitioning from atmospheric models directly (see Sect. 2.6). Noah-MP simulates canopy interception and throughfall of rain and snow, where the intercepted rain and snow on the canopy can go through unloading/dripping, frost, sublimation, melting, and freezing processes. Net evaporation loss from the canopy-intercepted liquid water (evaporation minus dew), net sublimation from the canopy-intercepted snow (sublimation minus frost), transpiration (via plant hydraulics), net soil surface evaporation, and net snowpack sublimation together contribute to the total surface ET. Noah-MP dynamically simulates multi-layer snowpack water storage (ice and liquid water) changes driven by snowfall/rainfall, frost, sublimation, freezing, and melting. The snowmelt water out of snowpack together with rainfall at the soil surface are further partitioned into surface runoff and infiltration based on multiple runoff and infiltration physics options (see Sect. 2.6). Soil moisture and





unsaturated water flow across soil layers are simulated using the one-dimensional Richards
equation. Two optional groundwater schemes, one without 2-D lateral flow (Niu et al., 2007) and
one with 2-D lateral flow (Fan et al., 2007; Miguez-Macho et al. 2007), are available in Noah-MP
to simulate groundwater dynamics, including groundwater recharge, water table change, baseflow,
seepage, and/or lateral flow. Noah-MP also includes dynamic irrigation and tile drainage processes
for agricultural management applications (Valayamkunnath et al., 2021, 2022). Figure 2
summarizes the key water processes and budget components as well as the water balance equation
in Noah-MP v5.0. Note that the water processes at glacier grids are treated similarly to those at
100% bare ground grids except that all the soil and subsurface hydrological processes are removed
and replaced by glacier ice (He et al., 2023).

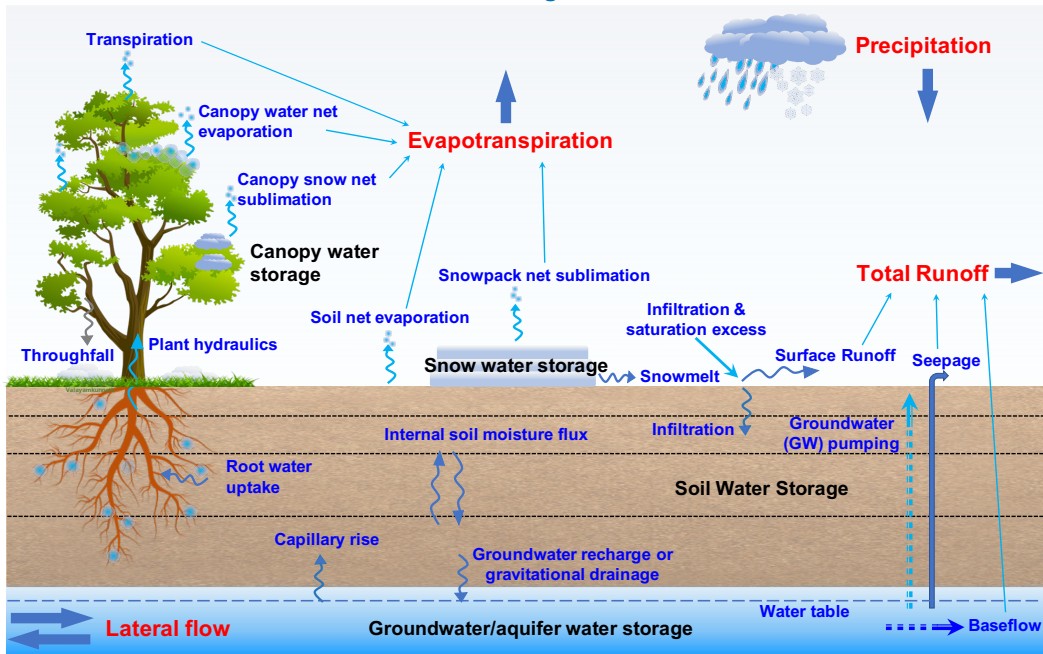

**Figure 2**. Schematic diagram of water budget and processes represented in Noah-MP version 5.0.
**2.4 Noah-MP biochemical processes**
Currently, the community version of Noah-MP only accounts for carbon processes for biochemical
cycles, while nitrogen dynamics and soil carbon dynamics have been developed in non-community
Noah-MP versions managed by individual research groups (e.g., Cai et al., 2016; X. Zhang et al.,



2022b). We will synthesize and integrate individual Noah-MP updates into the community version
in the future (see Sect. 2.5 for more discussions). Noah-MP simulates carbon processes for both
natural/generic vegetation (Niu et al., 2011) and explicit agricultural crops (Liu et al., 2016). The
carbon processes related to vegetation growth dynamics include (1) carbon assimilation from
photosynthesis by shaded and sunlit leaves, (2) carbon allocation to different parts of vegetation
(leaf, stem, wood and root) and soil carbon pools (fast and slow carbon), (3) carbon loss due to
respiration of different vegetation and soil carbon pools, (4) carbon transfer between vegetation
and fast soil carbon pools through vegetation (leaf, stem, wood and root) turnover and seasonal
death of leaf and stem, and (5) soil carbon pool conversion through soil carbon stabilization. The
total carbon flux to the atmosphere and net primary productivity are computed based on the
aforementioned carbon processes. Figure 3 summarizes the key carbon processes and budget
components as well as the carbon balance equation in Noah-MP v5.0. Note that the carbon
processes for crop growth are treated similarly to those of natural vegetation, except that the wood
component of plants is removed and the grain component of crops is added with additional carbon
conversion from leaf, stem, and root to grain depending on crop growing stages.

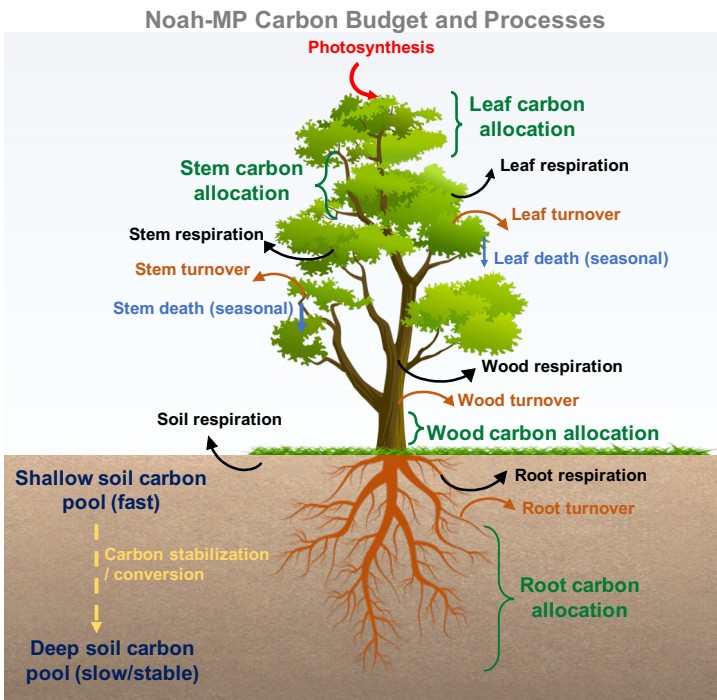

**Figure 3**. Schematic diagram of carbon budget and processes represented in Noah-MP version 5.0.





**2.5 Noah-MP physics updates since original development**

Since the release of the original Noah-MP in year 2011 (Niu et al., 2011), there are several important updates in Noah-MP physics. Some of the updates have been included in the community version of Noah-MP v5.0, while some are only available in the non-community versions managed by individual research groups. We will make efforts to synthesize and integrate individual Noah-MP updates into the community version in the future by working with those developer teams. Here, to the best of our knowledge, we briefly list the major Noah-MP physics updates from the community in the past decade.

The new/enhanced physics included in the community Noah-MP version 5.0 since 2011 are: (1) the Miguez-Macho-Fan (MMF) groundwater scheme (Barlage et al., 2015); (2) three additional runoff schemes: the Variable infiltration capacity (VIC), dynamic VIC, and Xinanjiang schemes (McDaniel et al., 2020); (3) tile drainage schemes (Valayamkunnath et al., 2022); (4) dynamic irrigation schemes (sprinkler, micro, and flooding irrigation) (Valayamkunnath et al., 2021); (5) a dynamic crop growth model for corn and soybean (Liu et al., 2016) with enhanced C3 and C4 crop parameters (Zhang et al., 2020); (6) coupling with urban canopy models (Xu et al., 2018; Salamanca et al., 2018) with local climate zone modeling capabilities (Zonato et al., 2021); (7) enhanced snow cover, snow compaction, and wind-canopy absorption parameters (He et al., 2021); (8) a wet-bulb temperature-based snow-rain partitioning scheme (Wang et al., 2019).

The new/enhanced physics currently not included in the community Noah-MP version 5.0 since 2011 are: (1) nitrogen dynamics (Cai et al., 2016); (2) big-tree plant hydraulics (Li et al., 2021); (3) dynamic root optimization (Wang et al. 2018) with an explicit representation of plant water storage (Niu et al., 2020); (4) additional snow cover parameterizations (Jiang et al., 2020); (5) coupling with a wind erosion model (Jiang et al., 2021); (6) a wetland representation and dynamics (Z. Zhang et al., 2022); (7) a unified turbulence parameterization throughout the canopy and roughness sublayer (Abolafia-Rosenzweig et al., 2021); (8) enhanced snow albedo representations (Abolafia-Rosenzweig et al., 2022b); (9) coupling with a snow radiative transfer (SNICAR) model (Wang et al., 2020, 2022); (10) an organic soil layer representation at forest floors (Chen et al., 2016) and a microbial-explicit soil organic carbon decomposition model (MESDM; X. Zhang et al., 2022b); (11) coupling with atmospheric dry deposition of air pollutant (Chang et al., 2022); (12) enhanced permafrost soil representations (X. Li et al., 2020); (13) spring wheat crop dynamics (Zhang et al., 2023); (14) new treatment of thermal roughness length (Chen and Zhang 2009); (15) the Gecros crop model (Ingwersen et al., 2018; Warrach-Sagi et al., 2022); (16) a 1-D dual-permeability flow model (based on the mixed-form Richards' equation) representing preferential flow through variably-saturated soil with surface ponding being developed in the University of Arizona.

**2.6 Noah-MP multi-physics options**






One unique feature and advantage of Noah-MP is the inclusion of multiple physics options for
different land processes for testing competing hypotheses (i.e., options) and multi-model ensemble
simulations. Table 1 summarizes all the available physics options in the community Noah-MP
v5.0. In particular, compared to previous Noah-MP versions, we have separated the runoff options
for surface and subsurface runoff processes, and added a new physics option for snow thermal
conductivity calculations, which were originally hard-coded without the namelist control
capability. More detailed descriptions of each physics option are provided in the technical
documentation (He et al., 2023).
**Table 1**. List of Noah-MP version 5.0 multi-physics options

| Noah-MP Physics | Option | Notes (* indicates the default option) |
|---|---|---|
| OptDynamicVeg<br><br>options for dynamic (prognostic) vegetation | 1 | off (use table LeafAreaIndex; use VegFrac = VegFracGreen from input) (Niu et al., 2011; Yang et al., 2011) |
| | 2 | on (together with OptStomataResistance = 1) (Dickinson et al., 1998; Niu and Yang, 2003) |
| | 3 | off (use table LeafAreaIndex; calculate VegFrac) |
| | 4* | off (use table LeafAreaIndex; use maximum vegetation fraction) |
| | 5 | on (use maximum vegetation fraction) |
| | 6 | on (use VegFrac = VegFracGreen from input) |
| | 7 | off (use input LeafAreaIndex; use VegFrac = VegFracGreen from input) |
| | 8 | off (use input LeafAreaIndex; calculate VegFrac) |
| | 9 | off (use input LeafAreaIndex; use maximum vegetation fraction) |
| OptRainSnowPartition<br><br>options for partitioning precipitation into rainfall & snowfall | 1* | Jordan (1991) scheme |
| | 2 | BATS: when TemperatureAirRefHeight < freezing point+2.2  (Yang and Dickinson, 1996) |
| | 3 | TemperatureAirRefHeight < freezing point (Niu et al., 2011) |
| | 4 | Use WRF microphysics output (Barlage et al., 2015) |
| | 5 | Use wet-bulb temperature (Wang et al., 2019) |
| OptSoilWaterTranspiration<br><br>options for soil moisture factor for stomatal resistance & ET | 1* | Noah (soil moisture) (Ek et al., 2003) |
| | 2 | CLM (matric potential) (Oleson et al., 2004) |
| | 3 | SSiB (matric potential) (Xue et al., 1991) |
| OptGroundResistanceEvap<br><br>options for ground resistent to evaporation/sublimation | 1* | Sakaguchi and Zeng (2009) scheme |
| | 2 | Sellers (1992) scheme |
| | 3 | adjusted Sellers (1992) for wet soil |
| | 4 | Sakaguchi and Zeng (2009) for non-snow; rsurf = rsurf_snow for snow (set in NoahmpTable.TBL) |
| OptSurfaceDrag<br><br>options for surface layer drag/exchange coefficient | 1* | Monin-Obukhov (M-O) Similarity Theory (Brutsaert, 1982) |
| | 2 | original Noah (Chen et al. 1997) |



| | | |
|---|---|---|
| OptStomataResistance<br><br>options for canopy stomatal resistance | 1* | Ball-Berry scheme (Ball et al., 1987; Bonan, 1996) |
| | 2 | Jarvis scheme (Jarvis, 1976) |
| OptSnowAlbedo<br><br>options for ground snow surface albedo | 1* | BATS snow albedo (Dickinson et al., 1993) |
| | 2 | CLASS snow albedo (Verseghy, 1991) |
| OptCanopyRadiationTransfer<br><br>options for canopy radiation transfer | 1 | modified two-stream (gap = $f$ (solar angle,3D structure, etc) < 1-VegFrac) (Niu and Yang, 2004) |
| | 2 | two-stream applied to grid-cell (gap=0) (Niu et al., 2011) |
| | 3* | two-stream applied to vegetated fraction (gap=1-VegFrac) (Dickinson, 1983; Sellers, 1985) |
| OptSnowSoilTempTime<br><br>options for snow/soil temperature time scheme (only layer 1) | 1* | semi-implicit; flux top boundary condition (Niu et al., 2011) |
| | 2 | full implicit (original Noah); temperature top boundary condition (Ek et al., 2003) |
| | 3 | same as 1, but snow cover for skin temperature calculation (Niu et al., 2011) |
| OptSnowThermConduct<br><br>options for snow thermal conductivity | 1* | Stieglitz scheme (Yen,1965) |
| | 2 | Anderson (1976) scheme |
| | 3 | Constant (Niu et al., 2011) |
| | 4 | Verseghy (1991) scheme |
| | 5 | Douvill scheme (Yen, 1981) |
| OptSoilTemperatureBottom<br><br>options for lower boundary condition of soil temperature | 1 | zero heat flux from bottom (DepthSoilTempBottom & TemperatureSoilBottom not used) (Niu et al., 2011) |
| | 2* | TemperatureSoilBottom at DepthSoilTempBottom (8m) read from a file (original Noah) (Ek et al., 2003) |
| OptSoilSupercoolWater<br><br>options for soil supercooled liquid water | 1* | No iteration (Niu and Yang, 2006) |
| | 2 | Koren's iteration (Koren et al., 1999) |
| OptRunoffSurface<br><br>options for surface runoff | 1 | TOPMODEL with groundwater (Niu et al., 2007) |
| | 2 | TOPMODEL with an equilibrium water table (Niu et al., 2005) |
| | 3* | Schaake scheme (original Noah) (Schaake et al., 1996) |
| | 4 | BATS surface and subsurface runoff (Yang and Dickinson, 1996) |
| | 5 | Miguez-Macho & Fan (MMF) groundwater scheme (Fan et al., 2007; Miguez-Macho et al. 2007) |
| | 6 | Variable Infiltration Capacity Model surface runoff scheme (Liang et al., 1994) |
| | 7 | Xinanjiang Infiltration and surface runoff scheme (Jayawardena and Zhou, 2000) |
| | 8 | Dynamic VIC surface runoff scheme (Liang and Xie, 2003) |





| | | |
|---|---|---|
| OptRunoffSubsurface <br><br> options for drainage & subsurface runoff | 1~8 | similar to runoff option, separated from original Noah-MP runoff option, currently tested & recommended the same option# as surface runoff (default) |
| OptSoilPermeabilityFrozen <br><br> options for frozen soil permeability | 1* | linear effects, more permeable (Niu and Yang, 2006) |
| | 2 | nonlinear effects, less permeable (Koren et al., 1999) |
| OptDynVicInfiltration <br><br> options for infiltration in dynamic VIC runoff scheme | 1* | Philip scheme (Liang and Xie, 2003) |
| | 2 | Green-Ampt scheme (Liang and Xie, 2003) |
| | 3 | Smith-Parlange scheme (Liang and Xie, 2003) |
| OptTileDrainage <br><br> options for tile drainage currently only tested & calibrated to work with runoff option=3 | 0* | No tile drainage |
| | 1 | on (simple scheme) (Valayamkunnath et al., 2022) |
| | 2 | on (Hooghoudt's scheme) (Valayamkunnath et al., 2022) |
| OptIrrigation <br><br> options for irrigation | 0* | No irrigation |
| | 1 | Irrigation on (Valayamkunnath et al., 2021) |
| | 2 | irrigation trigger based on crop season planting and harvesting dates (Valayamkunnath et al., 2021) |
| | 3 | irrigation trigger based on LeafAreaIndex threshold (Valayamkunnath et al., 2021) |
| OptIrrigationMethod <br><br> options for irrigation method, only works when OptIrrigation > 0 | 0* | method based on geo_em fractions |
| | 1 | sprinkler method (Valayamkunnath et al., 2021) |
| | 2 | micro/drip irrigation (Valayamkunnath et al., 2021) |
| | 3 | surface flooding (Valayamkunnath et al., 2021) |
| OptCropModel <br><br> options for crop model | 0* | No crop model |
| | 1 | Liu, et al. (2016) crop scheme |
| OptSoilProperty <br><br> options for defining soil properties | 1* | use input dominant soil texture |
| | 2 | use input soil texture that varies with depth |
| | 3 | use soil composition (sand, clay, orgm) and pedotransfer function |
| | 4 | use input soil properties |
| OptPedotransfer <br><br> options for pedotransfer functions, only works when OptSoilProperty=3 | 1* | Saxton and Rawls (2006) scheme |
| OptGlacierTreatment <br><br> options for glacier treatment | 1* | include phase change of glacier ice |
| | 2 | Glacier ice treatment more like original Noah |



## 3. Enhanced model modularization in Noah-MP version 5.0


In the Noah-MP v5.0, we have modularized all model physics by separating and re-organizing each code subroutine into individual process-level Fortran module file with new descriptive, self-





explanatory module and subroutine names. As such, each model physics or scheme has its own
separate module. Figure 4 shows the calling tree of the modularized Noah-MP main model physics
workflow. Figures 5-7 show the calling tree of the modularized energy, water, and carbon
processes, respectively. Compared to the previous Noah-MP versions that have a single lengthy
source file lumping together all model subroutines with non-self-explanatory names, the highly-
modularized model structure of the Noah-MP v5.0 provides a much more clear, neat, and
organized way for users and developers to understand and follow the model logics and physics.
These new modules use consistent coding format and standards, offering convenience for code
reading, writing, and debugging. The highly-modularized model structure also allows external
community weather/climate/hydrology models to easily adopt specific Noah-MP physical
processes/schemes as independent process-level module files and implement them for testing and
coupling.



**Figure 4**. The modularized Noah-MP main physics calling tree in version 5.0. Blue boxes indicate
water processes, orange boxes indicate energy processes, and green boxes indicate biochemical
processes. The direction of arrows indicates processes calling sequence and information flow. Note
that the 1-D glacier column model has similar structures as the main non-glacier model, except
that the vegetation-related processes are removed and soil is replaced by glacier ice.




**Figure 5**. The modularized Noah-MP energy processes calling tree in version 5.0. Note that the glacier model has similar structures except that the vegetation-related processes are removed and soil is replaced by glacier ice.



**Figure 6**. The modularized Noah-MP water processes calling tree in version 5.0. Note that the glacier model has similar structures except that it only includes the snowpack processes and soil is replaced by glacier ice.



328

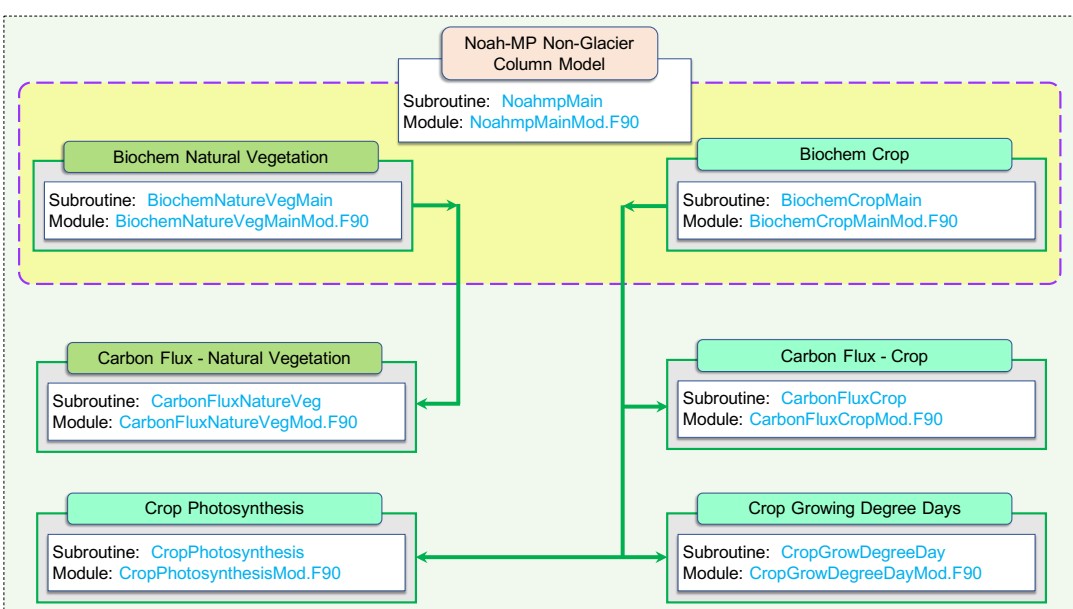

329

**Figure 7.** The modularized Noah-MP biochemical processes calling tree in version 5.0. Note that currently the Noah-MP v5.0 only includes carbon processes. Note that the CropPhotosynthesis module is not used currently to avoid inconsistency with the photosynthesis calculations from the canopy stomatal resistance module.

## 4. Enhanced data structure in Noah-MP version 5.0

In the Noah-MP v5.0, we have enhanced data structure with new hierarchical data types, which allows a more efficient and convenient control of model variables and substantially simplifies code structures and calling interface (Section 5). Figure 8 summarizes the new Noah-MP data type hierarchy and gives some examples of model variable expression based on the hierarchical data types. Specifically, we have defined an overarching "noahmp" main data type, which includes "forcing" for atmospheric forcing variable type, "config" for model configuration variable type with "domain" and "namelist" subtypes, "energy" for energy-related variable type, "water" for water-related variable type, and "biochem" for biochemistry-related variable type. The "energy", "water", and "biochem" types are further divided into "flux", "state", and "param" subtypes for flux, state, and parameter variables. This hierarchical data structure provides a better organization and management of model variables and their physical attributes. We have also optimized the variable declaration and initialization structures based on those new data types and consistent coding format and standard. In addition, we have re-defined many key local model state, flux, and parameter variables in the base code to be global variables in the refactored code, which allows a




better track and management of these variables for diagnosis, transfer between Noah-MP and host
models, and coupling with data assimilation systems.

(a)

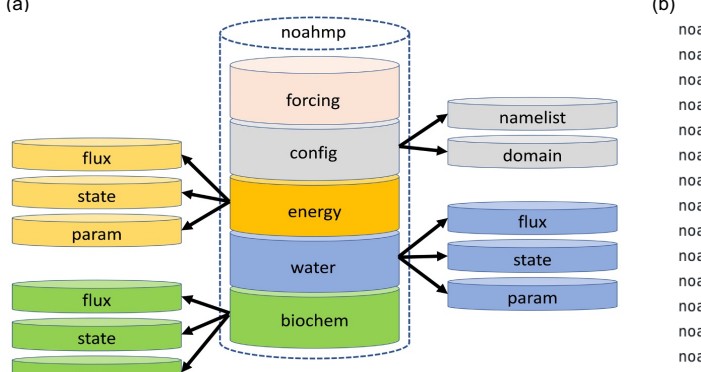

(b)

```
noahmp%forcing%PressureAirRefHeight
noahmp%forcing%RadLwDownRefHeight
noahmp%forcing%RadSwDownRefHeight
noahmp%config%nmlist%OptSnowSoilTempTime
noahmp%config%domain%FlagCropland
noahmp%config%domain%FlagSoilProcess
noahmp%config%domain%NumSoilTimeStep
noahmp%config%domain%SoilTimeStep
noahmp%water%param%IrriFracThreshold
noahmp%water%state%IrrigationFracGrid
noahmp%energy%state%LeafAreaIndEff
noahmp%energy%state%StemAreaIndEff
noahmp%energy%state%VegFrac
noahmp%energy%flux%HeatLatentIrriEvap
noahmp%energy%flux%HeatPrecipAdvCanopy
```

**Figure 8**. (a) The new hierarchical "noahmp" data types in the Noah-MP version 5.0. (b) Examples
of model variable expression using the hierarchical data types.

**5. Enhanced code structure in Noah-MP version 5.0**

Leveraging the model modularization (Section 3) and new data types (Section 4) in the Noah-MP
v5.0, we have further refined the code structure and subroutine interface. A graphical
representation of the refactored Noah-MP subroutine interface is depicted in Figure 9. Specifically,
the refined subroutine interface only requires passing the "noahmp" data type instead of each
individual variable names, because all relevant variables are defined and included in the "noahmp"
data type. This significantly simplifies the code structure with much more concise and neat
subroutine calls. The refined subroutine interface also makes future model development and code
changes simpler, more efficient, and less error-prone. For instance, if users want to add/remove a
variable for a specific physical scheme, they only need to edit as few as 3 module files: variable
type definition module, variable initialization module, the target physical scheme module, and if
needed, the variable input/output module. There is no need to go through and change all the
subroutine calls and interfaces that use the target variable.



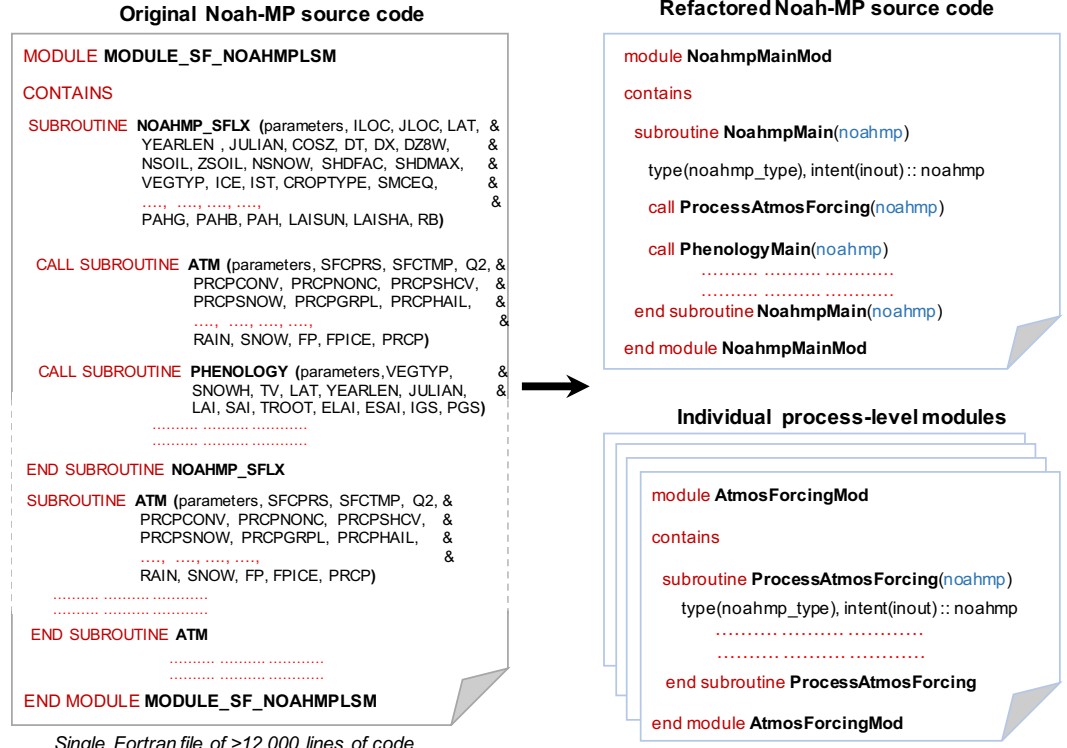

**Figure 9**. Demonstration of refactored subroutine interface and code structure in the Noah-MP version 5.0.

## 6. Enhanced variable naming in Noah-MP version 5.0

In the Noah-MP v5.0, we have also renamed all the model variables using a more descriptive and self-explanatory naming standard, which clarifies the physical meaning of variables directly by their names and hence substantially lowers the hurdles of reading and understanding the code and model physics. The original variable names in the previous Noah-MP versions are hard to understand, in which case users have to check back and forth the variable definition to know their physical meaning. For instance, the original variable name for canopy intercepted total water is "CMC", while the new name is "CanopyTotalWater". Table 2 gives more examples of the enhanced variable naming in Noah-MP v5.0. A detailed Noah-MP variable glossary listing variables' original and new names, physical meaning, data type, and unit is provided in the technical documentation (He et al., 2023) and the community Noah-MP GitHub repository.





**Table 2**. Examples of new variable names based on a more descriptive and self-explanatory
naming standard in the Noah-MP version 5.0, compared with the original names.

| Variable physical meaning/definition | New name | Original name | Variable Type | Unit |
|---|---|---|---|---|
| wetted or snowed fraction of canopy | CanopyWetFrac | FWET | Real | - |
| canopy intercepted liquid water | CanopyLiqWater | CANLIQ | Real | mm |
| canopy intercepted ice | CanopyIce | CANICE | Real | mm |
| canopy intercepted total water | CanopyTotalWater | CMC | Real | mm |
| canopy capacity for snow interception | CanopyIceMax | MAXSNO | Real | mm |
| canopy capacity for liquid water interception | CanopyLiqWaterMax | MAXLIQ | Real | mm |
| ice fraction in snow layers | SnowIceFrac | FICE_SNOW | Real | - |
| bulk density of snowfall | SnowfallDensity | BDFALL | Real | $kg/m^3$ |
| snow cover fraction | SnowCoverFrac | FSNO | Real | - |
| snow layer ice | SnowIce | SNICE | Real | mm |
| snow layer liquid water | SnowLiqWater | SNLIQ | Real | mm |


**7. Enhanced coupling structure with host models in Noah-MP version 5.0**

We have further updated the Noah-MP driver and interface coupled with potential host
weather/climate/hydrology models. Figure 10 summarizes the interface and coupling structures in
the Noah-MP v5.0. Specifically, the coupling interface includes: (1) defining a 2-D (for structured
grid mesh) or vectorized (for unstructured grid mesh) Noah-MP input/output data type
"NoahmpIO" to facilitate the input/output communication between host models and the core
Noah-MP 1-D column model ("noahmp" data type); (2) the initialization of the "NoahmpIO"
variables with values from host models; (3) the main Noah-MP driver that calls the core 1-D
column model and transfers between the "NoahmpIO" and "noahmp" variables as part of
input/output processes. Currently, the coupling of the Noah-MP v5.0 with the NCAR/HRLDAS
system has been successfully completed. The coupling of Noah-MP v5.0 with the NASA/LIS
system and the WRF-Hydro/NWM system is on-going. We also plan to couple the Noah-MP v5.0
with other host models in the future (Section 9), such as WRF, MPAS, and NOAA/UFS. Because
of the enhanced coupling interface and structure in Noah-MP v5.0, we will only need to slightly
adapt the coupling interface and driver to allow it to work with different host models. We will
manage and maintain the interface and driver code for each host model in the community Noah-
MP GitHub repository to ensure the compatibility between host models and updated core Noah-
MP source code in the future, which will allow smooth transition and seamless synthesizing of
Noah-MP updates in host models.



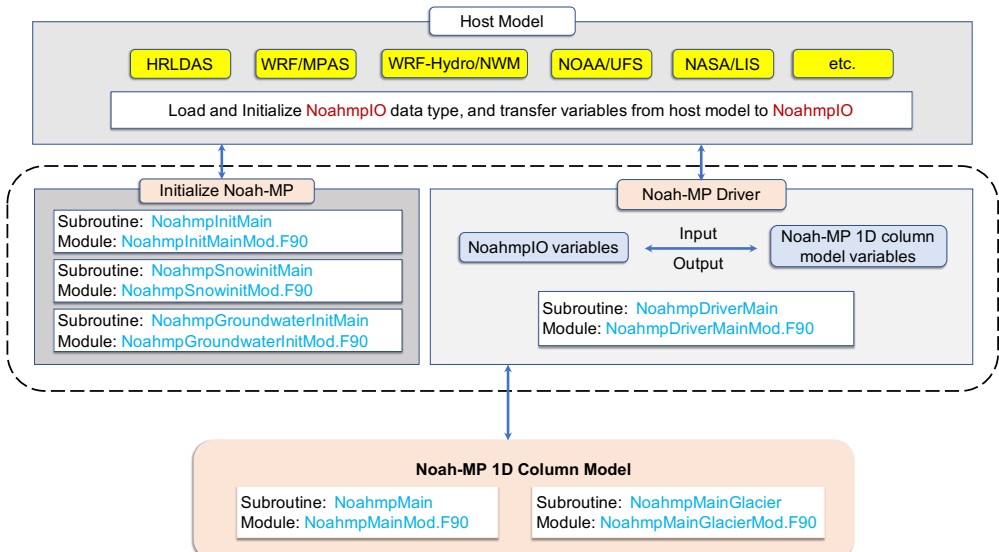

**Figure 10**. Workflow of the Noah-MP v5.0 driver and interface structures to couple with various host weather/climate/hydrology models.

## 8. Benchmarking for Noah-MP version 5.0

To benchmark the functionality, reproducibility, and computational efficiency of the modernized Noah-MP code, we have conducted a series of hierarchical test simulations during the course of Noah-MP refactoring. Specifically, after refactoring each major Noah-MP model component/physics (e.g., water, energy, carbon, etc.) listed in Figure 4, we built simple driver modules to conduct benchmark simulations using each of these model component/physics to test and ensure the bit-for-bit consistency between the refactored code and base code for all Noah-MP physics options. Here is an example for the refactored Noah-MP water component model we built for benchmarking during the course of refactoring: https://github.com/cenlinhe/NoahMP_refactor/tree/water_refactor, which was used to test the bit-for-bit consistency between the refactored and base Noah-MP water component codes.

After we completed the entire model refactoring, we have conducted another set of test simulations using the completed Noah-MP v5.0 to ensure its bit-for-bit consistency with the base model code for all different combinations of physics options as well as to benchmark its computational efficiency. These tests were conducted via 1-year point-scale SNOTEL 804-site simulations, 1-year 12-km gridded continental US simulations, and 1-year 1-km gridded simulations over central US agricultural regions (particularly to test individual and combination of physics options related to crop, irrigation, tile drainage, and groundwater). The tests all showed exactly the same results between the refactored and base simulations, with similar computational efficiency.


In addition, in order to provide the community with reference Noah-MP v5.0 model datasets for
future comparison and assessment, we have conducted 3 sets of benchmark simulations, including
21-year (2000-2020) 12-km continental US simulations driven by the NLDAS-2 atmospheric
forcings (Xia et al., 2012), 10-year (2009-2018) point-scale SNOTEL 804-site simulations over
the western US driven by observed precipitation and temperature as well as other NLDAS-2
atmospheric forcings downscaled to 90-m spatial resolution (He et al., 2021), and 1-year (2000)
4-km dynamic crop simulations over the U.S. Corn Belt region driven by the convection-
permitting WRF modeling (Zhang et al., 2020). We have archived all the atmospheric forcing
datasets, model setup input datasets, and model output datasets for these benchmark simulations.
Figure 11 shows an example of the model output. Note that a comprehensive evaluation of the
simulation results is outside the scope of this model description paper and will be done in the next
step.

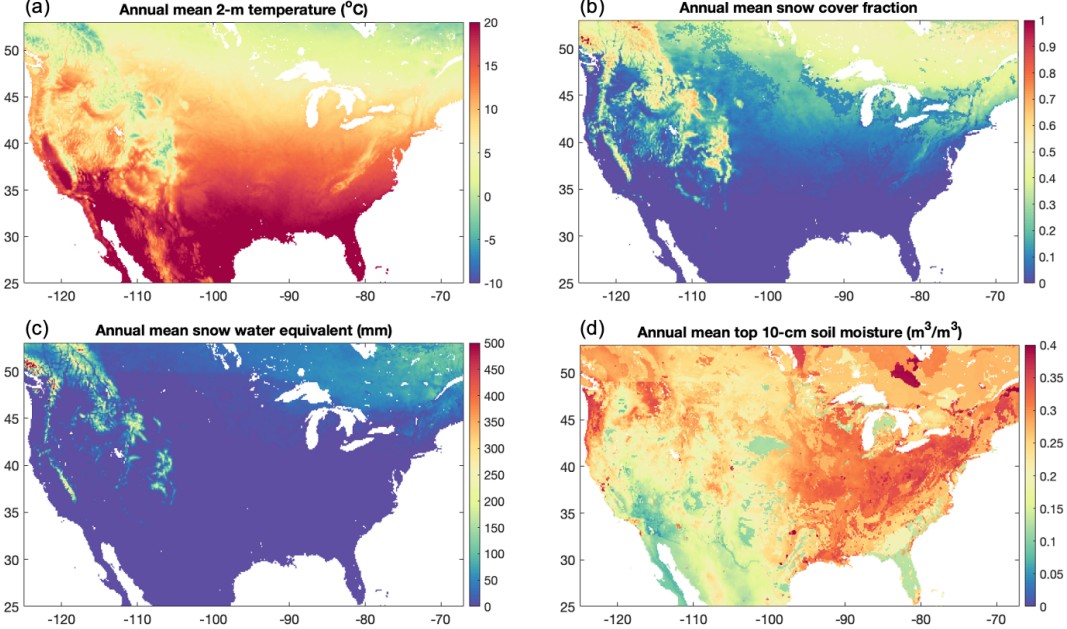

**Figure 11**. Demonstration of 20-year (2001-2020) annual mean (a) 2-m temperature, (b) snow
cover fraction, (c) snow water equivalent, and (d) top 10-cm soil moisture from the Noah-MP
version 5.0 12-km continental US benchmark simulations driven by the NLDAS-2 atmospheric
forcings.

**9. Model code and technical documentation for Noah-MP version 5.0**

We archive, manage, and maintain the Noah-MP v5.0 (together with previous code versions) at
the NCAR community Noah-MP GitHub repository (https://github.com/NCAR/noahmp) for



public access. We have also created a comprehensive technical documentation (He et al., 2023)
for the Noah-MP v5.0, available at http://dx.doi.org/10.5065/ew8g-yr95, which provides detailed
descriptions of model physics and formulations.
**10. Conclusions and future plans**
In this study, we modernized the widely-used state-of-the-art Noah-MP LSM by adopting modern
Fortran code and data structures and standards, which substantially enhances the model modularity,
interoperability, and applicability. The modernized Noah-MP has been released as the model
version 5.0, which includes the following key features: (1) enhanced modularization and
interoperability by re-organizing model physics into individual process-level Fortran module files,
(2) enhanced data structure with new hierarchical data types and optimized variable declaration
and initialization structures, (3) enhanced code structure and calling workflow by leveraging the
new data structure and modularization, (4) enhanced (descriptive and self-explanatory) model
variable naming standard, and (5) enhanced driver and interface structure to couple with host
weather/climate/hydrology models. The base code used for modernization is the Noah-MP version
4.5 (released in December 2022), and the modernization effort does not change model physics. In
addition, we have created a comprehensive technical documentation (He et al., 2023) of the Noah-
MP v5.0, and a set of benchmark simulation datasets. The Noah-MP v5.0 has been coupled to the
NCAR/HRLDAS system. Currently, the work of coupling the Noah-MP v5.0 with the latest
NASA/LIS system and the WRF-Hydro/NWM system is on-going. In the future, we also plan to
couple the Noah-MP v5.0 to other weather and climate models, including WRF, MPAS, and
NOAA/UFS. Overall, the modernized open-source community Noah-MP model will allow a more
efficient and convenient process for future model developments and applications.
**Code and data availability**
The Noah-MP model code is available at https://github.com/NCAR/noahmp
The coupled HRLDAS/Noah-MP model code is available at https://github.com/NCAR/hrldas
The Noah-MP technical documentation is available at http://dx.doi.org/10.5065/ew8g-yr95
The benchmark datasets can be provided by the corresponding author upon request, due to the
extremely large data size (8.8 TB).
**Author contribution**
CH, PV, and MB led the code refactoring effort with the help from all the other coauthors (FC,
DG, RC, GN, ZY, DN, ME, TS, RR). CH and PV led the technical documentation writing effort
with the help from all the other coauthors (MB, FC, DG, RC, GN, ZY, DN, ME, TS, RR). CH
conducted the benchmark model simulations. CH drafted the manuscript with improvements from
all the other coauthors (PV, FC, MB, DG, RC, GN, ZY, DN, ME, TS, RR).





**Competing interests**

The authors declare that they have no conflict of interest.



**Acknowledgements**

We thank Zhe Zhang (NCAR) and Ronnie Abolafia-Rosenzweig (NCAR) for helping with model code testing and for helpful discussions. We also acknowledge the strong support from the entire Noah-MP community. This study was supported by the US Geological Survey (USGS) Water Mission Area's Integrated Water Prediction Program, NOAA's Climate Program Office's Modeling, Analysis, Predictions, and Projections Program (MAPP), and the NCAR Water System Program. National Center for Atmospheric Research (NCAR) is a major facility sponsored by the National Science Foundation (NSF) under Cooperative Agreement #1852977. Any opinions, findings, conclusions, or recommendations expressed in this publication are those of the authors and do not necessarily reflect the views of the National Science Foundation.

527

528

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
