# Peer review of "Modernizing the open-source community Noah-MP land surface model (version 5.0) with enhanced modularity, interoperability, and applicability"

_EGUsphere, 2023_

## Author Comment (AC1)

[revised manuscript text omitted]

**Original Noah-MP source code**

```
MODULE MODULE_SF_NOAHMPLSM

CONTAINS

SUBROUTINE NOAHMP_SFLX (parameters, ILOC, JLOC, LAT, &
            YEARLEN , JULIAN, COSZ, DT, DX, DZ8W,      &
            NSOIL, ZSOIL, NSNOW, SHDFAC, SHDMAX,       &
            VEGTYP, ICE, IST, CROPTYPE, SMCEQ,         &
            ...., ...., ...., ....,                    &
            PAHG, PAHB, PAH, LAISUN, LAISHA, RB)

    CALL SUBROUTINE ATM (parameters, SFCPRS, SFCTMP, Q2, &
            PRCPCONV, PRCPNONC, PRCPSHCV,    &
            PRCPSNOW, PRCPGRPL, PRCPHAIL,    &
            ...., ...., ...., ....,          &
            RAIN, SNOW, FP, FPICE, PRCP)

    CALL SUBROUTINE PHENOLOGY (parameters,VEGTYP,      &
            SNOWH, TV, LAT, YEARLEN, JULIAN,   &
            LAI, SAI, TROOT, ELAI, ESAI, IGS, PGS)
            .......... .......... ...........
            .......... .......... ...........

END SUBROUTINE NOAHMP_SFLX

SUBROUTINE ATM (parameters, SFCPRS, SFCTMP, Q2, &
            PRCPCONV, PRCPNONC, PRCPSHCV,  &
            PRCPSNOW, PRCPGRPL, PRCPHAIL,  &
            ...., ...., ...., ....,        &
            RAIN, SNOW, FP, FPICE, PRCP)
            ......... ......... ...........
            ......... ......... ...........
    END SUBROUTINE ATM
            ......... ......... ...........
            ......... ......... ...........

END MODULE MODULE_SF_NOAHMPLSM
```
*Single Fortran file of >12,000 lines of code*

**Refactored Noah-MP source code**

```
module NoahmpMainMod contains

  subroutine NoahmpMain(noahmp)

    type(noahmp_type), intent(inout) :: noahmp

    call ProcessAtmosForcing(noahmp)

    call PhenologyMain(noahmp)
    .......... .......... ............
    .......... .......... ............
  end subroutine NoahmpMain(noahmp)

end module NoahmpMainMod
```

**Individual process-level modules**

```
module AtmosForcingMod contains

  subroutine ProcessAtmosForcing(noahmp)
    type(noahmp_type), intent(inout) :: noahmp
    .......... .......... ............
    .......... .......... ............
  end subroutine ProcessAtmosForcing end module AtmosForcingMod
```

[revised manuscript text omitted]

---

## Author Comment (AC2)

Dear Reviewer,

Thank you for the constructive comments. We have carefully studied your comments and carried out revisions accordingly. Below is a point-by-point response (marked as red) to the review comments. We hope you find our responses adequately address your comments and the revisions acceptable.

Sincerely,

Cenlin He (on behalf of all co-authors)

**Reviewer #1:**

This manuscript describes the modernization and refactoring of the widely-used state-of-art Noah-MP LSM, which was released as Noah-MP v5.0. The mordent Fortran code and data structures and standards are adopted in the refactoring. Five key features, including re-organized model physics inro individual process-level Fortran module files, enhanced data structure, enhanced code structure, self-explanatory variable naming standard, and enhanced interface structure to couple with the host models are introduced. It is introduced that the latest released Noah-MP has been coupled with NCAR/HRLDAS system. Some benchmark simulation results over the CONUS are presented. The work of coupling the Noah-MP v5.0 with the latest NASA/LIS system and the WRF-Hydro/NWM system is on-going. In the future, it is also plan to couple to other weather and climate models.

The original code of the model is lengthy single Fortran file. One has to read through the whole file and locate places interested. The refactoring procedures provide a modernized and interoperated model system, with which users could easily understand, modify or utilize in wide applications. As an open-sourced model, the simulation results should be reproducible with given forcing datasets. It is a substantial contribution to modelling science within the scope of Geoscientific Model Development. The manuscript is logical clearly, concisely, and well-structurally interpreted. In addition, model refactoring is a burdensome but less productive task. Not much people have willingness to do this work. This manuscript describes the refactoring conception and processes of the widely-used land surface model. For this kind fundamental work, encourage and circulate should be deserved. I think it is acceptable with minor revisions.

Response: We thank the reviewer for the positive comments. We have provided a point-by-point response below. The page and line numbers in our responses are referring to the track-change version of the revised manuscript.

The manuscript detailed describes the model structures, and what have been done with the model coupling to HRLDAS. My only concern is about the future plans. It is stated that the Noah-MP v5,0 will be coupled with more host models. It is encouraging. However, every model has its discrepancies. What are the advantages and disadvantages of this model? how about the future plans for model developments and applications? What is the next step in next several years to promote the advantages and makeup the weakness?

Response: Thanks for the constructive comments on the future plans.
(1) We agree that every model has its own disadvantages and advantages, including Noah-MP.

As we stated in the manuscript, the advantages of Noah-MP include (a) a comprehensive treatment of the coupled vegetation-snow-soil-hydrology system, which captures their complex interactions; (b) capabilities for additional critical land processes (e.g., crop, irrigation, tile drainage, groundwater, urban, and carbon cycles); (c) the inclusion of multiple physics options for different land processes, which allows the multi-physics model ensemble experiments for uncertainty assessment and testing competing hypotheses; (d) broad model applicability that allows to be used for various spatial and temporal scales as well as various application cases (e.g., weather prediction, climate projection, data assimilation, extreme weather/climate, hydrology, and agriculture); (e) high computational accuracy and efficiency that leads to the use of Noah-MP in many research and operational coupled modeling systems (e.g., NWM, UFS, LIS, WRF, MPAS).

The disadvantages of Noah-MP include the uncertainties and biases in some processes such as snowpack physics, vegetation dynamics, plant hydraulics, soil infiltration, and coupled carbon-nitrogen cycle, which have been pointed out by some previous studies cited in our manuscript, as well as some missing land surface processes such as accurate representations of blowing snow, wetland, wildfire disturbance, vegetation recovery and replacement. We expect these missing processes will be included in the model gradually in the future. For example, in our on-going work, we are implementing the wildfire disturbance process into Noah-MP.

(2) There are a few future plans for Noah-MP developments and applications based on the recent Noah-MP international workshop (https://ral.ucar.edu/events/2023/noah-mp-annual-users-workshop). Thus, we added the future plans to the revised manuscript in Section 10. "Conclusions and future plans" as follows:

"*The future plans for Noah-MP developments and applications include but not limited to (1) coupling with other widely-used weather/climate models (e.g., WRF, MPAS, NOAA/UFS), (2) enhancing capability of land data assimilation with Noah-MP, (3) enhancing plant hydraulics and soil hydraulics/hydrology schemes, (4) improving accuracy of applications in subseasonal-to-seasonal (S2S) forecasts, food-water security, and extreme weather/climate (e.g., fire, drought, flood, and heatwave), (5) including automated model parameter calibration/optimization algorithms, (6) enhancing modeling capabilities for rapid landscape transformation (e.g., deforestation/reforestation) as well as vegetation recovery and replacement after environmental disturbance, (7) including human management modeling (e.g., groundwater pumping), (8) including interactions with air pollution (e.g., pollutants' deposition and ozone damage to vegetation), (9) enhancing representation of subgrid heterogeneity, (10) improving high-resolution input datasets (e.g., soil properties and groundwater-related inputs), (11) creating a set of packages for code benchmarking and testing, model diagnostic, and better debugging capability.*"

(3) In the next few years, we plan to promote the Noah-MP advantages by enhancing its process representations and application in research and operational coupled weather/hydrology/climate modeling through coupling with various host models, and by enhancing its offline applications particularly in handling weather/climate extremes. In order to mitigate Noah-MP weaknesses, we plan to enhance the representations of key model processes and include missing physics as summarized above.

---

## Author Comment (AC3)

**A Point-by-Point Response to Comments of Reviewer #2**

Dear Reviewer,

Thank you for the constructive comments. We have carefully studied your comments and carried out revisions accordingly. Below is a point-by-point response (marked as red) to the review comments. We hope you find our responses adequately address your comments and the revisions acceptable.

Sincerely,

Cenlin He (on behalf of all co-authors)

**Reviewer #2**:

This paper describes the refactored version of the NoahMP land surface model. As the authors acknowledge, work described here is a software exercise and I do not find much utility in this manuscript, particularly given that a technical report (He et al. 2023) has already been developed. In my opinion, while the software exercise mentioned here sounds like a good step, it doesn't rise to the level of a paper for a number of reasons.

Response: We thank the reviewer for the comments, which help clarify the objectives of this paper. However, we do not agree with the reviewer on the statement that this work is not suitable for a paper. Instead, we believe it is very important to publish this refactoring work as an official citable paper in GMD for several reasons:

   1. Our model description manuscript is a perfect fit into the GMD journal scope, which is intended to recognize and promote the effort of model development. The overarching goal of GMD is to publish papers on the description, development, and evaluation of numerical models of the Earth system and its components, which is exactly the purpose of this manuscript to document a major refactoring effort of a widely used land model. As highlighted by the first reviewer, this refactoring work is a substantial contribution to modelling science and this burdensome but fundamental model development effort deserves encouragement and circulation via the journal publication like GMD. There are many similar model description papers published in GMD.

   2. We would like to clarify that this manuscript is totally different from the technical note (He et al., 2023). The technical note describes detailed physical parameterizations, mathematical formulation, and instructions on model input, setup, and configurations, whereas this manuscript here reports *the motivation and concept for refactoring Noah-MP*, refactored model structures, and key new features and differences from previous model versions. Thus, these two documents are structurally different but complimentary and serve to different purposes.

   3. This manuscript has important utilities, which helps project managers, scientists, model users and developers at large quickly understand the major changes of model code and data structures in this new version 5.0. This manuscript also provides a review of the past model physics

advances as well as key peer-reviewed reference for future studies on Noah-MP model applications and developments, which represents overall directions not only for Noah-MP but also other LSMs.

4. There is a unanimous support of this refactoring work from the entire Noah-MP community. We have presented this work in both AMS annual conference and the recent international Noah-MP workshop with more than 200 worldwide attendees, which received extremely positive feedback and support even from other modeling communities. In fact, a lot of users have already started to use the refactored Noah-MP version 5.0 since its release in April 2023, and these users require a citable peer-reviewed paper. One good example is that this new version 5.0 has already been implemented into the operational Korean Integrated Model system after about two months of the Noah-MP v5.0 release. This again highlights the importance of having a peer-reviewed citable publication as an official reference for this work.

For the detailed comments, we have provided a point-by-point response below. The page and line numbers in our responses are referring to the track-change version of the revised manuscript.

MAJOR COMMENTS

1. From what I can make out based on the description and the expertise of the authors, I think they missed a major opportunity in this refactorization exercise, as there is no mention or adherence to any formal software engineering principles. In many places (e.g. line 92), the mention of 'modern Fortran code standard' has been highlighted. Is there a formal standard that you are referring to? Be specific.

Response: This is a good point and thank you for the suggestion. Our model refactoring mainly follows the modern Fortran 2003 code standard (https://j3-fortran.org/doc/year/04/04-007.pdf). We have included this clarification throughout the revised manuscript where the "modern Fortran code standard" is mentioned. For example, in Line 92: we revised it to *"… follow the modern Fortran 2003 code standard (https://j3-fortran.org/doc/year/04/04-007.pdf)*."

2. Page 3: The five key features mentioned seem quite interrelated and qualitative. What does enhanced data structure, enhanced code structure, optimized variable declaration etc mean? What formal software engineering principles were followed? Did this effort involve software engineers? How do you know if this new version is 'enhanced'? Did you solicit the input of the community in designing this new framework?

Response: Thank you for the comments.
(1) The enhanced data structure means the new data structures now use hierarchical data types that which allows a more efficient and convenient control of model variables and substantially simplifies code structures and calling interface. This has been described in the manuscript in Section 4.

The enhanced modularization means highly modularized model physics with separate module files for each physical process, which facilitates future model development by allowing specific sub land-physics model to work in isolation, along with another sub model, or replaced without interfering with other parts of the model code. This modularization also allows other models to easily adopt specific Noah-MP physical processes/schemes as independent process-level

module files and implement them for testing and coupling. This has been described in the manuscript in Section 3.

The enhanced code structure means the much simplified subroutine calling interface and more concise calling workflow, which makes future model development and code changes simpler, more efficient, and less error-prone. This has been described in the manuscript in Section 5.

The optimized variable declaration means the new variable names that are more descriptive and self-explanatory with replacement of difficult-to-understand acronyms, particularly for non-native-English speakers. Also, the parameterized variable types are now used to make it easier to change the base precision (single or double) of numerical values. This has been described in the manuscript in Section 6.

The enhanced coupling structure means much simplified Noah-MP driver code and subroutine calling interface between the host weather/climate models and Noah-MP, which allows more efficient coupling coding work.

(2) This work followed the formal software engineering principles including using the principles of object-oriented design, separation of concerns, and data locality.

(3) Yes, this effort intimately involved software engineers. In fact, one of our coauthors, Ryan Cabell, is a very experienced scientific model software engineer in our institution and provides invaluable suggestions regarding the best soft engineering practices, which is an essential part of this refactoring effort.

(4) This new version is "enhanced" because it includes all the aforementioned software engineering enhancements in various model aspects, including the improved code and data structures, modularization, variable declaration, and coupling structure/interface.

(5) Yes, we have been holding monthly meetings with Noah-MP developers and users, where we invited input and feedback from the Noah-MP community in this refactoring effort. In fact, there is a unanimous support of this work from the Noah-MP community. We have presented this work in both AMS annual conference and international Noah-MP workshop, which received extremely positive feedback and support.

3. Section 3: Other than separating the code into different modules, has the calling structure changed? From a debugging point of view, having the source code in different modules is no different from having them in a single file. For example, are there explicit code locations of extensibility defined in the new structure?

Response: Yes, the calling structure has been completely changed, which is now more concise, transparent, easy to comprehend, and simplified by leveraging the modularization and new hierarchical data types. Also, very lengthy subroutines have been broken into more specific parts corresponding to specific physical process. This will make debugging easier and provide more granular locations for code replacement/extensions and interfacing with model coupling libraries such as BMI or NUOPC.

4. One of the major impediments to using land surface models is the specification of input datasets. How does the new structure provide an improvement in this regard?

Response: This refactoring work re-organizes hierarchical data structures and allows a broad range of existing and future land-specific datasets to be loaded more efficiently.

5. Finally, this refactorization (despite the wide use of Noah-MP) seems like an insular effort. Did you solicit the requirements of different communities that use NoahMP (NWP, Hydrology, Data Assimilation, Crop modelers, etc.)?  Did you ensure that this new infrastructure is actually more efficient for including in systems like NWM, WRF, LIS, etc.?

Response: Thank you for the comments.
(1) Yes, we received input and feedback from the entire Noah-MP community including various Noah-MP developers and users related to NWP, hydrology, data assimilation, etc. There is a unanimous support of this refactoring work from the Noah-MP community. Two co-authors of this paper (Mike Barlage and David Gochis) are intimately involved in real-time operational NWP (NOAA UFS) and hydrologic predications (National Water Model and WRF-Hydro). We have presented this work in both AMS annual conference and international Noah-MP workshop, which received extremely positive feedback and support. In fact, a lot of users have already started to use the refactored Noah-MP version 5.0 since its release in April 2023. One good example is that this new version 5.0 has already been implemented into the operational Korean Integrated Model system and being implemented into the WRF-Hydro/NWM.

(2) The enhanced modularization and new data and code structures provide a more granular structure and a more concise calling interface and workflow, which makes it easier to integrate the entire Noah-MP or subsets of Noah-MP physics with host models particularly those already having similar refactored structures as what Noah-MP does. The coupling practice we are currently doing for LIS/NoahMP and WRF-Hydro/Noah-MP confirms a more efficient implementation of this new refactored Noah-MP version.

MINOR COMMENTS
1. Line 26 and 28: The modularization, interoperability mentions are repeated. Additionally, it is mentioned again in the text, and in Conclusions. Please reduce the redundancy of such descriptions.

Response: As suggested, we have removed the "interoperability" in Line 28 and main text including the conclusion section to reduce the redundancy.

2. Line 46: 'Modern LSMs have been …. as indispensable components …'

Response: Fixed as suggested.

3. Page 2, last para: This is an impressive set of references of the use of NoahMP. I still think it could use some improvements; Liu et al. 2017 is not provided. There are several use of NoahMP for data assimilation than mentioned here, please include them.

Response: As suggested, we have included a few additional references about data assimilation in the last paragraph of Page 2 as well as providing Liu et al. (2017) in the reference list.

4. Line 97: This is not a 'study', rather a description of the software reorganization of the code.

Response: We have revised "study" to "effort".

5. Page 4, paragraph beginning on line 140: I suggest rewriting this para to improve the readability. In this para, there are four sentences that begin with the style of 'Noah-MP does this'.

Response: We have revised this paragraph to improve the readability as follows:
*"The Noah-MP land grid is divided into two sub-grid tiles, namely vegetated and non-vegetated grounds, based on vegetation cover fraction. The biogeophysical and biogeochemical processes are treated separately for the vegetated and bare grounds. A "big-leaf" canopy treatment is adopted, which is characterized by canopy properties dependent on vegetation types. Noah-MP accounts for a multiple-layer snowpack, where snow ice and liquid water content, density, depth, and temperature are simulated dynamically. There are also multi-layer soil thermal and hydrological processes with dynamically evolving soil temperature and water content. The vegetation, snow, and soil components in Noah-MP are closely coupled and interacted with each other via complex energy, water, and biochemical processes."*

6. Line 263: What is the point of describing new physics that is NOT included in the community release? Is it listing different physics options being worked on? Please explain the significance or remove this para.

Response: The purpose of summarizing the model advances that is not in the community version is to give readers a complete picture of the Noah-MP development since its original release. Note that this is the first paper including such summary for previous Noah-MP developments. More importantly, we have plans to integrate these individual Noah-MP updates into the community version as new physics options in the future by working with those developer teams. In addition, users who are interested in leveraging those existing model capabilities that are not in the community version could reach out to the original developers through the reference list in this paragraph. Overall, we believe this paragraph is important and decided to keep it.

7. Section 4: What is the definition of flux, state, and parameters?

Response: The flux is the rate of transfer of energy, water, or carbon through a surface, such as sensible heat flux, evaporation water flux, or carbon photosynthesis flux. The state is the state variable for energy, water, or carbon, such as temperature, soil moisture, or leaf carbon storage. The parameter is the variable prescribed in the lookup table or code for use in physical parameterizations, such as soil heat capacity, saturated soil hydraulic conductivity, or reference carbon assimilation rate.

8. Figure 11: 2m temperature is an input to the model. Why is that being benchmarked?

Response: All the atmospheric forcing (including surface temperature) is input to Noah-MP at a specified height (typically 10 meter), and then Noah-MP calculates the 2-m temperature for vegetated and bare portions of each grid based on land surface energy balance. Finally, the grid-mean 2-m temperature is a weighted average of the 2-m temperature for vegetated and bare portions of each grid. Thus, the resulting 2-m temperature will not be exactly the same as the input surface temperature forcing. That is why it is included as an output in our benchmark datasets.